# Probing with Each Step: How a Walking Crab-like Robot Classifies Buried Cylinders in Sand with Hall-Effect Sensors

**DOI:** 10.3390/s24051579

**Published:** 2024-02-29

**Authors:** John Grezmak, Kathryn A. Daltorio

**Affiliations:** Department of Mechanical and Aerospace Engineering, Case Western Reserve University, Cleveland, OH 44106, USA; kati@case.edu

**Keywords:** unexploded ordnance detection, Hall-effect sensing, soft sensors, marine robotics, convolutional neural networks

## Abstract

Shallow underwater environments around the world are contaminated with unexploded ordnances (UXOs). Current state-of-the-art methods for UXO detection and localization use remote sensing systems. Furthermore, human divers are often tasked with confirming UXO existence and retrieval which poses health and safety hazards. In this paper, we describe the application of a crab robot with leg-embedded Hall effect-based sensors to detect and distinguish between UXOs and non-magnetic objects partially buried in sand. The sensors consist of Hall-effect magnetometers and permanent magnets embedded in load bearing compliant segments. The magnetometers are sensitive to magnetic objects in close proximity to the legs and their movement relative to embedded magnets, allowing for both proximity and force-related feedback in dynamically obtained measurements. A dataset of three-axis measurements is collected as the robot steps near and over different UXOs and UXO-like objects, and a convolutional neural network is trained on time domain inputs and evaluated by 5-fold cross validation. Additionally, we propose a novel method for interpreting the importance of measurements in the time domain for the trained classifier. The results demonstrate the potential for accurate and efficient UXO and non-UXO discrimination in the field.

## 1. Introduction

Unexploded ordnances (UXOs) are known to contaminate many shallow water environments around the world. For example, more than 10 million acres of U.S. coastal and territorial waters have been identified as potentially contaminated with UXOs [1]. Over time, these UXOs can become obstructed or partially obstructed by natural or human-made phenomena, lowering their visibility and raising the risk of accidental detonation. This poses a major safety hazard to industrial developers, commercial fishing and transportation systems as well as the general public seeking to utilize these areas. Consequently, there is a need for UXO detection and localization strategies that are accurate, efficient and safe.

The vast majority of state-of-the-art UXO detection and localization strategies involve remote sensing modalities. These include acoustic scattering measurements (e.g., sidescan or multi-beam synthetic aperture sonar (SAS)) [2,3,4], magnetometry measurements [5,6,7,8], electromagnetic imaging (EMI) sensors [9] or optical measurements (LiDAR, video) [10], and while remote sensing can be preferable to contact or close proximity sensing to avoid accidental detonation, it is often susceptible to environmental noise when the geographical range of sensing is large, or when water conditions are turbid. Additionally, depending on the surveying range, some modalities such as magnetometry may lack classification ability [6], requiring follow-up surveys of identified areas of interest with a finer sensing range and potentially human divers for final verification and disposal. Thus, alternative strategies which can accurately classify and discriminate UXOs while lowering the need for additional costly interventions should be explored.

In this paper, we propose the use of a legged crab-like robot with Hall effect-based sensors for UXO and non-UXO discrimination. In previous work, we have shown that hexapod robots with crab-inspired limbs are suitable for navigation in shallow, turbid water [11], making them suitable platforms for incorporating underwater UXO sensing strategies. We have also shown that Hall-effect magnetometers measuring the field of a magnet embedded in compliant robot limbs can be used to classify the terrain from measurements taken as the robot dynamically interacts with the terrain [12]. In this study, we modify these Hall-effect sensors for the purpose of UXO and non-UXO discrimination. By choosing magnetometers with resolution small enough to detect weak external magnetic fields but still larger than Earth’s field, measurements are a function of load and proximity to magnetic objects, which is expected to provide valuable discriminatory feedback on the object type and orientation as the robot interacts with the object.

To validate the use of the proposed sensing modality for UXO and non-UXO discrimination, we collect a dataset of measurements collected as a robot equipped with Hall-effect sensors walks over several UXO and UXO-like objects that are half-buried in dry sand. We specifically focus on a single set of conditions (i.e., object depth and orientation, robot walking speed, soil properties) in order to evaluate the discriminatory power of the proposed method when the only variables are the magnetic properties of the encountered objects. A convolutional neural network (CNN) is trained on inputs based on the three-axis measurements collected in the time domain and is validated using 5-fold cross validation. To explore the importance of individual measurements on the decisions of the trained classifier, we further propose a neural network (NN) interpretation method based on a modified occlusion sensitivity analysis method which highlights the time-based importance of individual measurements. We use this method to discover the physical events represented by the measurements that have discriminatory power for distinguishing between UXO and non-UXO objects.

## 2. Related Works

In this section, we review state-of-the-art underwater UXO detection and discrimination strategies based on the more commonly used magnetometry and acoustic-based sensors and discuss their limitations for robust classification. We also review robotic classification of objects in granular media.

### 2.1. UXO Detection and Discrimination

#### 2.1.1. Magnetometry

Magnetometers as a UXO sensing modality typically measure the total magnetic field or the difference(s) between two or more physically separated magnetometers (i.e., gradiometer configuration) with the goal of detecting magnetic objects as distortions in the Earth’s magnetic field [6] and have typically been deployed on towed vessels [8] or UAVs [13]. Tensor gradiometer configurations are common as they can directly determine the location of a magnetic object as well as the magnitude and orientation of their magnetic moments [7,8]. Seidel et al. [13] successfully demonstrated UXO detection in known munitions dumpsites with a gradiometer configuration mounted on the nose of a hovering UAV.

Magnetometry-based solutions have the advantage of being well suited for buried UXOs since the obscuring sediment layer typically does not interfere with their magnetic signatures. However, the main drawback is their general lack of classification ability, due to the nature of the measurements and the requirement of a large amount of good training data for classification algorithms [6]. Recently though, some efforts have focused on mitigating this issue. For example, the feasibility of UXO and non-UXO discrimination from magnetic data using a probabilistic framework has recently been investigated by Wigh et al. [14], and Blachnik et al. [15] proposed a numerical modeling method for generating training data based on the Digital Twin concept.

#### 2.1.2. Acoustic Sensing

Acoustic methods are typically based on measurements from sidescan or downward-facing SAS mounted on ships. Some research efforts have been made to model the acoustic response of underwater UXO and UXO-like objects [3,16]. For example, Kargl et al. [3] conducted a series of monostatic and bistatic acoustic scattering measurements using SAS to investigate their discrimination abilities for underwater UXOs. To address the issue of the large data volume required to cover the many possibilities of UXO orientation and submersion depth, several studies have used physics-based modeling techniques to generate synthetic data [16,17]. For example, Bucaro et al. [18] used finite element methods to generate simulated scattering measurements of buried UXOs and UXO-like objects at various orientations. Recently, some studies have investigated machine learning (ML) methods for automatic object classification from measurements [4,17,19,20]. Hall et al. [17] generated a dataset of acoustic color data from simulated SAS measurements using the fast ray model to train a matched subspace classifier. Hoang et al. [4] proposed a physics-inspired neural network classifier whose architecture can capture relevant physical phenomena inherent in SAS measurements and showed that it outperforms other state-of-the-art algorithms on real datasets when given limited training data.

Acoustic methods can achieve large detection ranges, especially with low-frequency sidescan SAS, which is beneficial for surveying large areas or hard to reach locations. However, besides also suffering from the requirement of large amounts of good training data for classification algorithms, there is a trade-off in the frequencies and incident angles used to excite potential UXOs, requiring broadband systems for balancing detection range and detection of buried objects [4].

### 2.2. Robotic Perception of Objects in Granular Media

Robotic tactile perception in granular media is complicated by the complex interactions of end effectors and the granular media, which exhibit varying physical properties that can make tactile sensing modalities difficult to interpret [21]. Robotic classification of objects in granular media through tactile sensing has received little attention. Jia et al. demonstrated tactile perception of objects in granular media using a teleoperated robotic finger equipped with a BioTac sensor [22]. The multimodal sensing (vibration, internal fluid pressure, fingerpad deformation) was shown to provide additional information important for classification compared to single modalities. This work was expanded upon to develop a framework for robotic tactile perception, mapping, and haptic exploration in granular media, which used a physics-based sensor model of the tactile measurements [23]. Using a similar approach, Patel et al. [24] demonstrated object identification in granular media using a robotic finger equipped with a modified GelSight sensor [25], a type of visuotactile sensor.

In this work, we propose the use of a legged robot equipped with Hall effect-based compliance sensors in the legs for detection of UXOs partially buried in granular media. This approach has the advantages over existing methods in that the sensing modality can be easily waterproofed and integrated into the existing robot design, preserving desired geometry. Additionally, as we have previously shown that the classification of the terrain properties using this sensor type can be achieved during locomotion [12], the proposed method can be used to detect buried UXOs during locomotion, enabling quicker coverage of target areas than methods requiring stationary robotic hands.

### 2.3. Underwater UXO Detection with Robotic Systems

Few studies have investigated the integration of underwater UXO detection sensor modalities into robotic systems. The UAV with a nose-mounted gradiometer in [13] was capable of detecting 81 mm UXO shells from a distance of 1 m from the sea floor. Harper and Dock [26] reported the use of a chemical olfactory-based sensor mounted on both a crawler and ROV robot. To the authors’ knowledge, this work is the first to investigate the integration of underwater UXO sensing into walking multi-legged robots.

## 3. Materials and Methods

### 3.1. Robotic Platform

A modified version of our custom Hexapod robot from our previous work, *Sebastian* [11,12], is chosen as the platform for evaluating the proposed UXO detection method and is shown in Figure 1. The robot uses Savox SW2210SG IP67 rated servo motors as actuators, which have been evaluated in previous field testing to remain operable after extensive use in shallow-water environments and a water-tight chassis for electronics. Since no additional waterproofing is required for deployment in shallow-water environments, the application of interest in this research, the robot serves as a suitable platform for evaluating the proposed method.

Each leg has two segments of lengths 75.1 mm and 120.6 mm (proximal and distal) for which the actuators have parallel axes of rotation, restricting the motion of an end effector to a plane. With two sets of three parallel legs mounted on opposite ends of the chassis, the robot can perform coordinated sideways gaits with arbitrary end effector paths. The Hall-effect sensors are integrated into the distal segments, which maintain the same crab-like geometry at the end effectors as in previous iterations. The geometry allows the end effectors to penetrate the granular media (up to approximately 60 mm) during walking, allowing the Hall-effect sensors to potentially directly interact with buried UXOs, and thus achieve more effective sensing. Together with the chosen limb lengths, the robot is capable of stepping over buried or partially-buried UXO-like objects with a diameter of 60 mm, the size of the inert UXO model and similarly sized objects of interest in this work.

### 3.2. Hall-Effect Sensor Development

The concept of using Hall-effect sensors to measure the local change in magnetic field due to magnets embedded in soft material has recently been increasingly used for novel designs of different types of robotic sensors, including tactile sensing fingertips [27,28,29], skin-like sensor arrays [30,31] and multi-axis load cells [32,33,34]. This transduction method requires fewer wires for communication and is straightforward to integrate into existing robotic mechanisms compared to other commonly used transduction methods, such as piezoresistive and optical-based sensors. To our knowledge, this and our previous work [12] are the first to utilize Hall-effect sensors for legged robot applications. However, the application in the current work is unique in that the Hall-effect sensors are intended to measure both the field changes from the locally embedded magnets as well as those from external objects to be classified.

#### 3.2.1. Choice of Hall-Effect Sensor

To realize the aforementioned capabilities, the Hall-effect sensor should have a sensing range with a lower limit small enough to capture the magnetic field of UXOs at distances expected during robot–object interaction (e.g., while walking over the object) while being large enough to filter out the effects of Earth’s magnetic field and potential ferromagnetic properties of soil that could be encountered. At the same time, the upper limit should ideally be large so that embedded magnets can be placed closer to the sensor without causing saturation, which is beneficial for increasing the measurement sensitivity to small, applied end-effector forces, yet it should be small enough to not impact sensitivity due to bit resolution restrictions. To this end, the TLV493D three-axis magnetometer is chosen for its 130 mT upper range and 12-bit data resolution, which makes it sensitive to displacements of small magnets when near the sensor chip while also being (1) sensitive to displacements near UXOs (e.g., the 57 mm M86 inert round used in this study) and (2) unaffected by rotations in Earth’s field. The sensor also is capable of a sampling rate of >30 Hz when used in conjunction with two additional sensors connected to a multiplexer (Adafruit TCA9548A), which allows the sensor to capture robot interaction dynamics with good time resolution. Additionally, this sensor was found in previous work to not be sensitive to ferrous material in the soil of a real beach environment testing site when used in a similar implementation as the current work.

#### 3.2.2. Sensor Design and Choice of Parameters

To integrate the Hall effect sensor into the distal segments of *Sebastian*, we make use of the design flexibility of mold elastomers and 3D printing. Specifically, the sensor and magnet are embedded within opposite ends of an elastomer layer that spans the entire cross section of a limb segment, as shown in Figure 2. For a sufficiently compliant elastomer, this will allow relative movement between the top and bottom elastomer layer when the load is applied to the end effector, which in turn creates a relative displacement of components mounted at these locations (i.e., the sensor and magnets) and, thus, a change in the local field at the location of the Hall-effect sensor.

To determine the elastomer layer thickness, we assume the use of a N52 4.7625 mm thickness and outer diameter neodymium magnet as the embedded magnet, which is of reasonable size and strength to produce measurable changes for small displacements at distances less than 1 cm from the TLV493D sensor. We also assume the use of Vytaflex 20 urethane rubber as the material, which was found to have an appropriate resulting Young’s Modulus for this application as well as good shear resistance when bonding to the aluminum bracket and PLA. The cured material also possesses water-resistant properties, enabling it to serve as an additional protective layer for encasing the Hall-effect sensor, guarding it against water damage. Modeling the layer as a simple rectangular beam structure and assuming a maximum normal and shear loading (applied at one end) of 20 N, we find a thickness of 5 mm provides adequate sensitivity.

#### 3.2.3. Sensor Fabrication

The process for fabricating a limb distal segment with integrated Hall-effect sensor is as follows. The upper portion of the segment consists of an aluminum bracket that attaches to the actuator. The center of this bracket is drilled to allow a PCB with the Hall-effect sensor to be mounted as shown in Figure 2c. The bottom portion is 3D printed with PLA material, with one end (the side attaching to the elastomer layer) having a cutout to house the magnet, and the other end serving as the end effector with the desired dactyl inspired geometry, as shown in Figure 2b. To bond the upper and lower portions via the elastomer layer, removable 3D printed molds are designed to hold the parts at a distance of 5 mm while creating a space for the mold rubber to be poured into, as shown in Figure 2d. All mold surfaces in this space are sprayed in the mold release before the rubber is poured to prevent it bonding to these surfaces. After the rubber is poured into the mold, it is allowed to cure for 24 h before removing the molds and applying load to the resulting sensor.

#### 3.2.4. Sensor Evaluation

A batch of six sensors were prepared using the above procedures and evaluated for their performance properties. Some variability was found in the no load measurements, likely due to individual variation of relative Hall-effect sensor and magnet distances due to imprecision in the 3D printed parts and part positioning during casting. However, each sensor was found to exhibit fairly linear responses to both horizontal and vertical forces applied to the end effectors, which suggests the no load variability can be accounted for easily with proper scaling. Additionally, negligible drift in the no load measurements were found when comparing their values before and after the experiments described below were carried out.

The sensitivity to the field of the M86 inert UXO was also evaluated. Each sensor was found to have changes in measurements of approximately 4 μT, 0.5 μT and 1 μT in the x, y and z directions, respectively, when the UXO was moved with the centerline level to the Hall-effect sensor to a distance of 1 cm from the front side of the sensor. These changes are an order of magnitude greater than the variability of measurements from applied loads, which suggests the resulting sensor measurements will be able to capture the effects of stepping nearby the inert UXO.

### 3.3. Time Domain Occlusion Sensitivity

In this work, we are interested in classifying samples of measurements obtained from the Hall-effect sensors as a robot interacts with an object. Artificial NNs are a suitable choice as a classifier to map the complex relationships between the measurements and their associated object class. However, one drawback of NNs is the difficulty in deriving insights into the decision-making process of trained NNs with large numbers of parameters, giving them a reputation as being a “black box” [35]. To address this issue, several NN interpretation methods which focus on determining the contribution of local features (e.g., the inputs) to the output of a particular NN decision have been developed [36]. However, these methods all consider the relative influence of the individual feature to a decision by considering the input as a whole. In this work, considering the time-based nature of the measurements, we are interested in determining the relative influence of a measurement on an NN output by treating it as additional temporal information available to the network. In other words, we are interested in how much a given input feature, ideally corresponding to a particular sensor measurement in time, changes the decision of the network with respect to the decision made based only on the features corresponding to previous measurements.

To achieve this goal, we choose a CNN architecture for the classifier for the following two reasons: (1) having been shown to be achieve accurate classification of inputs representing time series [37], CNNs can relax the demand for signal preprocessing before being input to the network; (2) while normally requiring an input of fixed length, CNNs can effectively take inputs of variable length through zero padding an input array to match the required shape. Accordingly, by using an array of time series measurements as an input to the CNN, each feature will be mapped to a particular measurement in time, giving each feature intuitive physical significance. Additionally, by leveraging the variable input length capability, the temporal significance of individual features as described above can be determined.

A visualization of the inputs used to compute the temporal significance of an individual feature using the proposed time domain occlusion sensitivity (TDOS) analysis is shown in Figure 3. A temporal significance value is assigned to a feature corresponding to
(1)$$S=f\left(x+\delta_{x}\right)-f\left(x\right),$$
where *S* is the temporal significance value, *f* is the function representing the classifier, *x* is an input with nonzero values only for the features preceding the feature of interest, and $\delta_{x}$ is an input where the only nonzero value is that of the feature of interest. If $\delta_{x}$ is assumed to be a small change in the input relative to *x*, then by approximating $f(x+\delta_{x})$ using a first-order Taylor expansion as
(2)$$f\left(x+\delta_{x}\right)\approx f\left(x\right)+{\left(\frac{\partial f}{\partial x}{|}_{x}\right)}^{T}\cdot \left(x+\delta_{x}-x\right),$$
the temporal significance is then approximated as
(3)$$S=f\left(x+\delta_{x}\right)-f\left(x\right)\approx {\left(\frac{\partial f}{\partial x}{|}_{x}\right)}^{T}\cdot \delta_{x}.$$

Thus, assuming the resulting function represented by the trained model is reasonably linear in the space defined by $\delta_{x}$, the significance value approximately represents the change in the model output (and, for the case of a CNN classifier model, the class probability) as a measurement defined by $\delta_{x}$ is obtained.

### 3.4. Experimental Setup

To validate the proposed sensing modality for UXO and non-UXO discrimination, an experiment is carried out in a laboratory setting to collect a dataset of measurements as the robot interacts with an inert UXO and UXO-like objects. A small container is filled with sand (Pavestone natural play sand) such that the depth is at least 20 mm greater than the vertical length of the dactyl (60 mm) in all areas to prevent the dactyls from touching the bottom surface. An inert M86 UXO with a maximum outer diameter of 57 mm is considered as a representative UXO for its relative size to the robotic platform. Two UXO-like objects are also chosen for evaluating the ability to distinguish between objects of similar size but with different magnetic properties: a physical model of the M86 UXO with similar inertial properties but different magnetic properties and a PVC pipe with 60 mm outer diameter with weights added to match the inertial properties of the M86. Thus, all objects considered in this case study differ only in their magnetic signature, which are expected to be discriminatory for their classification. Six different object classes are considered for data collection, which are shown in Figure 4. The six classes with associated labels are (1) no object (None), (2) simulated UXO (SUXO), (3) inert UXO (IUXO), (4) PVC pipe (PVC), (5) inert UXO adjacent to PVC pipe (IUXO-PVC), and (6) PVC pipe adjacent to inert UXO (PVC-IUXO). The order of the labels in the final two classes indicate the order in which the robot is designed to walk over them (e.g., IUXO-PVC indicates the robot walks over the inert UXO before then walking over the PVC pipe adjacent to the inert UXO).

For a each trial of measurement collection, the object or objects are partially buried in the sand such that the sand surface level is aligned with the midline of the objects. To interact with the objects, the robot uses a tripod gait with phases denoted by the gait sequence diagram in Figure 5. The robot starts with the two outer front legs, F1 and F3, positioned against one side of the object (or first object in the case of dual object classes). The tripod formed by F1, F3 and the rear middle leg R2 then performs a raise–swing–plant sequence to lift and plant F1 and F3 over the object, while the legs forming the opposing tripod move in a stance phase to move the body forward. A designed stance and swing length of 100 mm and swing height of 80 mm are determined empirically as adequate values for allowing the legs to complete the sequence, ending with the dactyl planting just beyond the object without any dactyl interference. The previous actions are then repeated with the roles of each tripod flipped, which allows the tripod formed by legs F2, R1 and R3 to step over the second object for the dual object classes.

For each object class, a total of 10 trials of measurements are performed for data collection, resulting in a dataset size of 60 samples. The order in which the measurements are taken for each class is chosen randomly, with 5 trials for one class taken before switching to another class. Between each trial, the sand is leveled such that the sand condition is approximately constant for each trial. During each trial, 3-axis Hall-effect sensor measurements are collected by the three front legs at a sampling rate of 30 Hz, which results in 120 measurements collected for each measurement axis.

### 3.5. Machine Learning

#### 3.5.1. Data Pre-Processing

Each trial of measurements provides 9 different streams of 120 sensor measurements. To make full use of the temporal information contained in these measurements, the raw signals of a trial are first each individually normalized using min-max normalization and subsequently concatenated into a 9 × 120 array. Here min-max normalization is used to preserve the “shape” of the measurements and hence the ability to interpret the physical meaning of the inputs. The 9 × 120 arrays are directly used as inputs to CNN models with no further pre-processing.

#### 3.5.2. Architecture Considerations for TDOS

In order for TDOS to produce significance values which can safely be assumed to represent the change in class probability with respect to the addition of a measurement, the resulting trained model needs to be reasonably linear in the locations of each $\delta_{x}$ as they are defined in Section 3.3. To encourage simplicity of the trained models, we use the following considerations in choosing the CNN architecture. First, excessive numbers of layers and kernels or neurons within a layer are avoided. This is achieved by lowering their values starting from arbitrary choices that produce perfect accuracy using standard model evaluation techniques (e.g., cross validation) until the accuracy begins to drop. Second, the rectified linear unit (ReLU) function is used as the activation function for each layer besides the output layer, which simplifies the network in that many of the neurons will be inactive for a given input due to the nature of this function. Third, bias terms are not included in the neuronal output computations in order to reduce the number of parameters in the network.

#### 3.5.3. Model Training

The architecture used for all CNNs trained in this work based on the above considerations is summarized in Table 1. The output layer uses softmax activation, and cross entropy is used as the loss function for training. To further encourage the simplicity in the trained model for use with TDOS, early stopping is used with a patience value of 20 and the maximum allowed number of training epochs of 150.

## 4. Results and Discussion

### 4.1. Time Domain Measurements

The raw data for the 10 trials completed for each object class are averaged in the time domain and is plotted in Figure 6. From this visualization, inter-object class and inter-leg differences in measurements can be easily observed. For example, an obvious difference can be seen in the measurements of the outer legs and middle leg in the time period extending slightly before and after time t = 1 s, where the left and right legs are in swing and the middle leg is rolling forwards towards the object during the stance. Specifically, while the measurements for the outer legs are stable during this period, the middle leg measurements have a noticeable spike (just prior to t = 1) followed by a decay when there is an object in the sand.

An interesting observation is that for the measurements corresponding to the None class, the spike happens in the opposite direction to that of the other object classes. A possible explanation for this is as follows. During the stance motion, the middle leg pulls the robot forward, with the dactyl orientation moving through an pre-determined angle range. In previous work, we describe this angle as the angle between the ground and dactyl (AGD) [38]. At the end of the stance, the dactyl will be pointed further away from the direction of travel, and the weight of the robot pushes the dactyl inwards toward the center of the robot. However, if there is an object, as the middle leg “rolls” onto the object, a greater portion of the weight is then supported by the leg segment instead of the dactyl, and the reaction force opposing the dactyl motion in the sand, which acts in the opposite direction to the weight load (i.e., away from the robot), becomes greater than the weight reaction force.

The outer leg measurements after time t = 2 s, during which these legs are planted and transition into the stance phase while the middle leg swings, show large inter-class differences as seen in different magnitudes of the several peaks occurring during this time. This indicates that the proposed sensing modality for UXO and non-UXO discrimination is capable of producing distinguishable signatures in the measurements that could aid in determining the underlying object class. However, it is still difficult to achieve this through visual inspection of the raw measurements alone, thus making an automated solution using CNNs more attractive.

### 4.2. Machine Learning Results

To evaluate the performance of a CNN classifier trained on the dataset according to the pipeline outlined above, 5-fold cross validation is performed 20 times to mitigate the effects of random chance due to the small dataset size, and the overall average of the average testing accuracy for each instance is obtained. Using this metric, an overall average of 97.7% is obtained, which demonstrates the promising potential for this sensing modality for UXO and non-UXO discrimination.

The trained CNNs obtained during model evaluation are further evaluated using TDOS. For a cross validation instance where each split resulted in 100% testing accuracy, the model trained using the fewest number of epochs due to early stopping is chosen for further evaluation using TDOS. For each sample in the original dataset, TDOS is applied to the full feature vector to obtain a significance value for each feature. The significance values are then plotted as a heat map where the color for a given feature corresponds to its relative significance compared to the full sample. A plot of these heatmaps for the entire dataset is shown in the top portion of Figure 7.

When comparing the heatmaps for samples within a given class, consistency in the locations of positive significance values (i.e., red areas) and negative significance values (i.e., blue areas) can be seen. This suggests that similar portions of these samples have similar impacts on the classification result of the CNN. Given the time-based nature of the inputs, this would suggest that the measurements collected during similar events in time (e.g., legs coming in contact with object during stance, dactyls penetrating sand near object, etc.) have similar effects on the classification result. Additionally, when comparing heatmaps for samples belonging to different classes, there are event-dependent differences in the significance value intensities and signs. For example, during the first stance forward event, the no object samples show negative or neutral values while other classes, such as the SUXO and dual object classes, show high intensity positive values. Similar observations can be made for the events corresponding to the first tripod planting and the second tripod moving in the stance phase. This suggests that certain events during the robot–object interaction can have varying significance with regards to the classification result depending on the underlying object class.

To further evaluate the time-based nature of the decisions made by the CNN, we also consider the evolution of the CNN outputs as the number of non-zero elements (i.e., the number of measurements remaining in the occluded sample) is increased. In particular, after computing the CNN outputs for each occlusion size for a given sample, the feature corresponding to the last point in time where the network decision (i.e., the output with the largest magnitude) switches from one class to another is recorded, as is the class from which the prediction switched. These features are indicated as points along the time axis, with the color indicating the class from which the prediction last switched from, in the bottom portion of Figure 7. It can be seen that for the majority of samples, the CNN output last switched from the None class, and there is variance between the classes in the times at which the last switch occurred. For example, for the PVC class samples, the last switch tends to occur in the time frame of planting the first tripod and lifting the second tripod, while for the SUXO and IUXO-PVC classes, the last switch tends to occur during the stance phase of the first tripod. These observations align with locations of high-intensity positive significance values in the corresponding heatmaps. However, the last switch information can be complementary to the significance values for understanding the physical reasoning behind the CNN’s decision. For example, together, these two evaluation techniques allow a user of this particular CNN model to know that the time frame of planting the first tripod and raising the second tripod is informative for the model to not only predict the class as PVC but also to distinguish between the classes PVC and None.

### 4.3. Effect of Occluded Training Samples

The effect of including time-occluded samples in the CNN training process is also investigated. For a maximum occlusion size of *N*, the process of generating the associated augmented dataset is as follows. For each sample, occluded versions of the sample are obtained for occlusion sizes of 0, 1, …, *N* − 1, *N* by multiplying it element-wise by a mask with values of 1 corresponding to locations where features are kept and 0 where features are occluded. For example, for occlusion size *N*, the mask has a value of 0 for the 9 × *N* features corresponding to the last *N* data points and 1 elsewhere. Each of these occluded samples are added to the dataset. Thus, for a given maximum occlusion size, the augmented dataset has a size of 60*(*N* + 1) samples. CNNs are trained on augmented datasets for maximum occlusion sizes of 20, 40, and 60 using the same pipeline as in the previous analysis. Again, the models are evaluated using 20 repetitions of 5-fold cross validation. A representative model for each maximum occlusion size is chosen by randomly selecting a cross-validation instance where the testing accuracy is 100% and choosing the model trained using the fewest number of epochs. A summary of the models for each maximum occlusion size is shown in Table 2. Each dataset with occluded samples achieves 100% average testing accuracy with models trained using less than the maximum allowed 150 epochs, indicating that augmenting the dataset in this way has a positive effect on model training accuracy and efficiency.

The models in Table 2 are also evaluated using TDOS and last switch information, the results of which are plotted in Figure 8. Considering individual datasets, similar observations about the heatmaps and last switch plots when compared to the non-augmented dataset can be made. Namely, there is intra-class similarity and inter-class variability in the heatmaps, and the last switch locations have inter-class variability as well. However, there is a noticeable trend as the maximum occlusion size increases in that the locations of high-intensity significance values and last switches occur at earlier points in time. For example, for the case of maximum occlusion size *N* = 60, most of the visible portion of the heatmap occurs within the time frame of the first half step, and all of the last switch points happen just before the outer legs begin planting at the end of swing. These results indicate that by augmenting the dataset using time-based occlusion, CNN models can learn to distinguish between the object classes using features corresponding to earlier events in time.

## 5. Conclusions

This paper presents a novel sensing modality for UXO and non-UXO discrimination, which uses Hall-effect sensors embedded in the legs of a legged robot capable of walking in granular media. An experiment is conducted to validate the efficacy of this sensing modality, in which a dataset of sensor measurements is collected by having a robot walk over six classes of UXO and UXO-like object configurations (including no object) that are half buried in sand. A CNN architecture is trained on this dataset and is demonstrated to obtain very good classification accuracy using a cross-validation metric. Specifically, it is shown that a CNN can achieve an average 97.7% accuracy for distinguishing between 60 samples representing the six considered object classes. By augmenting the dataset by including partially time occluded versions of the original 60 samples, the CNN can achieve an average of 100% accuracy. We also introduce a novel method for determining the significance of the input features used for classification with respect to their temporal information, which we call TDOS. It is shown through heatmap visualization of these significance values that there is intra-class similarities and inter-class differences, suggesting the measurements collected during different events in time can be relevant for different object classes. Furthermore, by determining which feature causes the network to switch its classification output, one can obtain a sense of how much of the sample is needed for the CNN to give an accurate prediction. In environments with UXOs, a wrong step has the potential to disrupt a previously stable environment, causing a detonation. So while our current work suggests that we can detect UXO-like objects by walking over them, these approaches will also provide baseline confidences that enable future real-time decisions, for example, to avoid planting a step or to place the legs carefully to grasp particular objects. Furthermore, visualization of these confidence changes will be helpful as sensors are added and combined to increase confidence more quickly in even more varied conditions.

### Limitations and Future Work

This work evaluated the proposed method under a specific set of conditions, namely, constant object size, depth and orientation, granular material properties and robot walking speed. This is ideal for evaluating the capability of the robotic system to obtain relevant information from the objects’ magnetic signatures and inertial properties during a walking motion, through an assessment of the trained classifier performance and visualization of measurement temporal significance through TDOS. However, real-life applications will undoubtedly have different conditions which may affect the ability of a classifier trained on the original conditions to discriminate the object from new measurements. For example, the depth of the objects in the granular media will affect which parts of the legs interact with the object and the intensity of the interaction, affecting the sensor feedback coming from end-effector deflection. This will also affect how close the Hall-effect sensors are positioned relative to the objects and different points of the nominal motion. Either effect could be enough to cause failure of a pre-trained model to accurately discriminate the object type. Media properties will also affect the measurements in the same way; less penetrable soil will affect both the interacting parts of the legs with the objects due to varying penetration depth of the end effectors, as well as the distance from Hall-effector sensor to objects. Finally, the parameters of the walking gait used (e.g., speed, step height, step length) will have similar effects on the classification ability of pre-trained models relying on time-domain measurements as inputs, as the time-dependent events important for discrimination could produce different measurement signatures when the speeds of the end effectors relative to the objects are changed.

Another limitation of this study is that while it addressed the case of distinguishing between a real inert M86 and similarly sized objects by a magnetic signature, other UXO models will have different yet magnetic signatures. This will give dependence of the classifier on having trained-on samples corresponding to the UXO model of interest. For UXOs much larger than those used in this study, the robot legs would have to be scaled up in order to achieve sufficient stepping height and distance in order to replicate the probing motion carried out in this study.

Future work will involve creating a more generalized model that can automatically take into account different environmental properties (e.g., material properties, ground slope) and operating conditions (e.g., robot gait parameters) when making a decision. For example, information about the changes in measurements due to these changes can be obtained through both first principles and additional dataset collection. These factors can then be estimated through information obtained from other available sensing modalities and transformed to a format usable by a detection algorithm. Future work will also involve additional use of probing motions after initial detection has been made to generate a more confident prediction of object type and properties, such as orientation and depth, to inform subsequent actions such as retrieval.

## Figures and Tables

**Figure 1 sensors-24-01579-f001:**
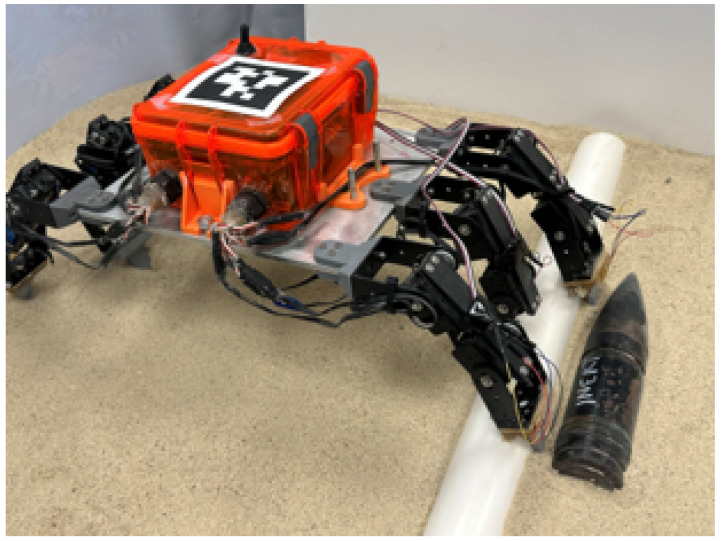
Hexapod robot *Sebastian* walking in sand. The three legs on the right of the picture are implemented with Hall-effect sensors, and are stepping between a PVC pipe (left) and inert UXO (left) of similar diameter.

**Figure 2 sensors-24-01579-f002:**
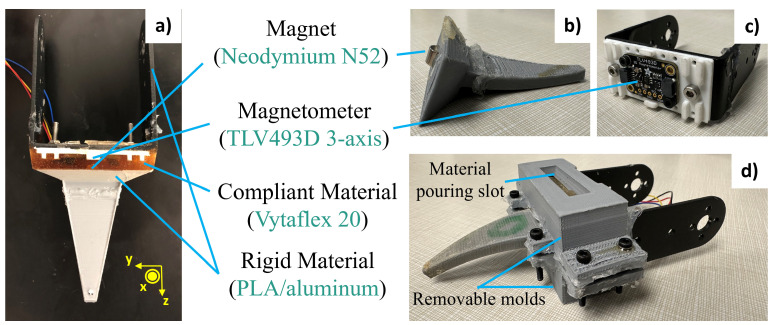
Hall-effect sensor for UXO detection embedded in leg segment. (**a**) Full assembly; measurement directions are denoted by the axes shown; (**b**) lower segment with embedded magnet; (**c**) upper segment with mounted Hall-effect sensor; (**d**) segments held apart 5 mm with removable molds.

**Figure 3 sensors-24-01579-f003:**
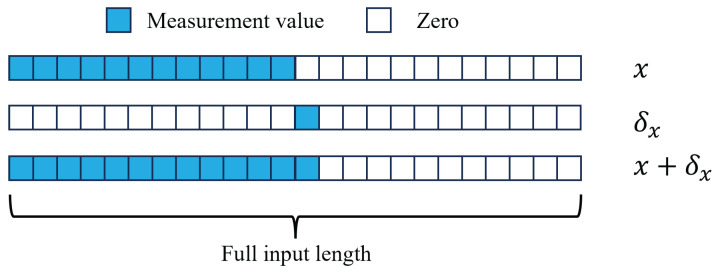
Definitions of inputs used for time domain occlusion sensitivity computation.

**Figure 4 sensors-24-01579-f004:**
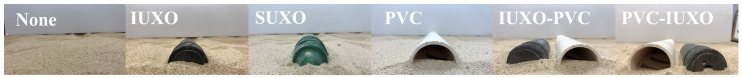
Object classes used in the experimental study. From left to right: no object (None), simulated UXO (SUXO), inert UXO (IUXO), PVC pipe (PVC), inert UXO next to PVC pipe (IUXO-PVC), PVC pipe next to inert UXO (PVC-IUXO).

**Figure 5 sensors-24-01579-f005:**
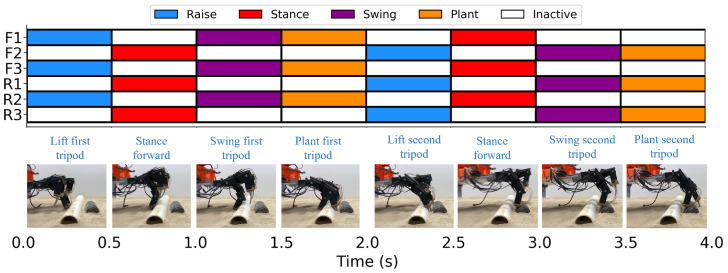
Gait-sequence diagram for the gait used by the robot for data collection.

**Figure 6 sensors-24-01579-f006:**
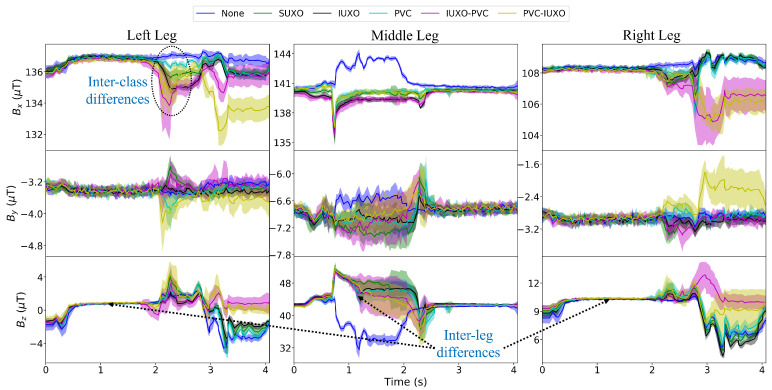
Averages of time-domain sensor measurements from each trial for the six object classes, shown as mean values (solid line) with standard deviations (shaded areas). Inter-class differences as seen by differences in the trends of the average values for each class in the measurements of a single leg are indicated for the left leg *x*-direction measurements. Inter-leg differences as seen by the overall trend differences in all classes at a given point in time are indicated for the *z*-direction measurements for each leg at approximately time *t* = 1 s.

**Figure 7 sensors-24-01579-f007:**
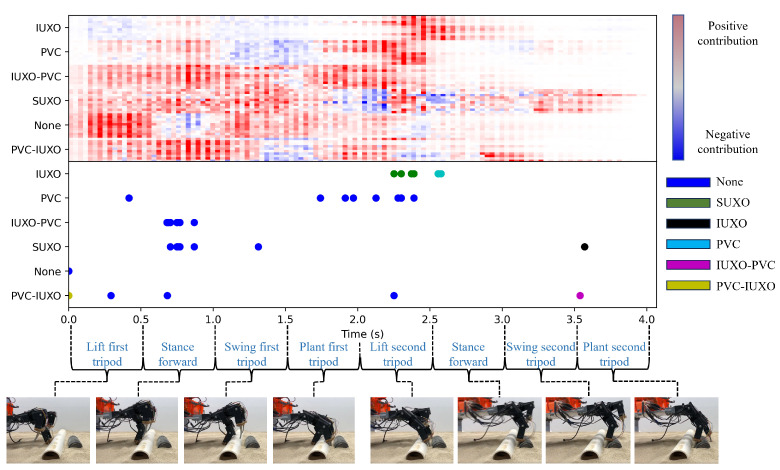
(**Top**) Heatmaps of significance values obtained from applying TDOS to each sample in the dataset. (**Bottom**) Locations of last prediction switching points in time, with color indicating the class from which the prediction switched from.

**Figure 8 sensors-24-01579-f008:**
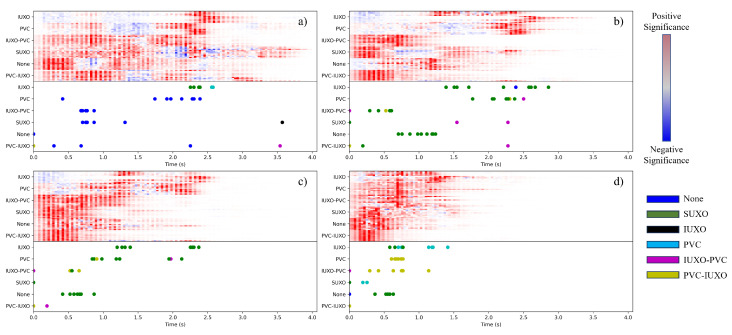
Heatmaps of TDOS analysis for CNNs trained with occlusion size of (**a**) 0, (**b**) 20, (**c**) 40, and (**d**) 60.

**Table 1 sensors-24-01579-t001:** CNN architecture used for all models.

	Input	Conv. Layer 1	Pooling Layer 1	Conv. Layer 2	Pooling Layer 2	Dense Layer	Output
Output size	9 × 120	9 × 118 × 10	9 × 59 × 10	9 × 55 × 12	9 × 27 × 12	32 × 1	6 × 1
Activation	None	ReLU	None	ReLU	None	ReLU	Softmax

**Table 2 sensors-24-01579-t002:** Summary of chosen models from training results on datasets with different maximum occlusion sizes.

Maximum Occlusions Size	Average Testing Accuracy	Epochs Trained
0	97.7%	150
20	100%	66
40	100%	45
60	100%	54

## Data Availability

The data presented in this study are openly available in Hall-Effect-Data at https://github.com/Crab-Lab-CWRU/Hall-Effect-Data.

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
