# Peer review of "Probing with Each Step: How a Walking Crab-like Robot Classifies Buried Cylinders in Sand with Hall-Effect Sensors"

_sensors, 2024, doi:10.3390/s24051579_

Round 1
Reviewer 1 Report
Comments and Suggestions for Authors
In the manuscript, the authors constructed a legged crab-like robot, building upon their prior research. They trained a convolutional neural network using collected data for validation and employed a neural network interpretation method to differentiate between UXO and non-UXO objects. The article is well-written, but additional information would be needed. Minor revision is suggested.
- In line 126, the authors described the robot as waterproof, but no data related to waterproofing is presented in the manuscript. Could the authors provide relevant information or data to support this claim?
- The robot is capable of stepping over objects with a diameter of 60 mm. Is this sufficient for real-world scenarios? If not, could the authors provide insights into how to increase the stepping distance?
- Can the robot maintain stability under different environmental conditions, such as changes in soil magnetic properties?
- Could the authors provide a summary of the limitations of their work, and suggest potential avenues for future research?
Reviewer 2 Report
Comments and Suggestions for Authors
Overall: The research content is quite intriguing. It involves the installation of Hall sensors on the joints of a robot's legs to control its movement around unexploded ordnance and dummy ordnance while collecting sensor data to form a dataset. Machine learning methods are then employed to differentiate between unexploded and dummy ordnance.
Detail problems:
- The introduction section should clarify why the research focuses on objects visible on the surface of sand. It should also address whether objects buried in the sand are within the scope of this study.
- More descriptions of the robot control system and experimental procedures should be included. Particularly, the capabilities of individual sensors shown in Figure 2 and the testing analysis should be elaborated. Additionally, details about the Hall sensor parameters, data collection range, frequency, and optimal distance or angle for obtaining better data should be provided. Furthermore, the description should explain how the robot interacts with unexploded ordnance, whether it crosses over or steps on it. If crossing over, details on how to accurately cross over, such as the distance between the two types of objects (IUXO-PVC and PVC-IUXO), should be addressed.
- A detailed explanation of the differences between UXO and non-UXO should be provided. Is it only the casing that differs?
- What impact would placing IUXO on top of PVC have on the results?
- In the conclusion section, it would be beneficial to list the research content and innovations in a clear and concise manner to enhance readability.
- Discussions on existing robots and how they are paired with detection instruments should be included.
- Why was only the scenario of unexploded ordnance partially buried in the sand with a 60mm diameter studied? How was this diameter determined, and why was only the scenario of partial burial in sand considered? The issue of a small dataset is also mentioned. Shouldn't the study expand the dataset by varying the burial depth, size of unexploded ordnance, and even variables such as how the robot passes over them and at what speed to enhance the machine learning dataset?
- Could the conclusion section include some quantitative descriptions? Additionally, it should address the limitations of this study and outline potential future work, especially regarding the study's boundaries, the problems that quantitative descriptions can solve, and those they cannot.
Reviewer 3 Report
Comments and Suggestions for Authors
The article talks about a methodology to detect UXOs using Hall effect sensors, in order to determine the values of said objects, CNNs are used. For this purpose, a sensor was designed whose main component is a hall effect sensor, which was located in the legs of the robotic platform designed for this purpose. Different experiments were carried out to validate the detection procedure of the various elements to be examined and an analysis is carried out where its validity and use in more critical environments can be determined.
There are some comments I want to share:
1. The manufacturing process is described broadly but at the level of the design is not seen, other than what is described below. It would be very valid for the replication of this method if this process were more technical and parts of its design were described or shared.
2. The platform used is referenced in two more articles. However, it is clear that the actuators used would not allow this device to be validated at depths greater than one meter where UXOs are generally found. It is possible that such modification alters the data obtained with the specified sensor. How could this process be improved or removed from the failure of changing these actuators?
3. When describing the use of the classifier, the designation of the inputs is not clear because it dedicates very little to the description of the possible occlusions and the strategy addressed. Can the detector itself solve the problem if the data is pre-processed? This is described in a paragraph in a section ahead where there is no clarity.
4. Figure 5 describes the gait sequence used for data collection. However, the movement carried out by the robot is not understandable.
5. A diagram of the architecture used for CNNs is needed
6. I consider that although there are different examples proposed for detection, different experiments should be considered in which detection is carried out with different conditions.
7. Figure 6 contains a lot of information that is not easily visible, I propose using another graphic methodology for its visualization.
Round 2
Reviewer 2 Report
Comments and Suggestions for Authors
The authors have addressed my concerns. I suggest acceptance
Reviewer 3 Report
Comments and Suggestions for Authors
I consider that the manuscript could be accepted and published